# A Myosin II-Based Nanomachine Devised for the Study of Ca^2+^-Dependent Mechanisms of Muscle Regulation

**DOI:** 10.3390/ijms21197372

**Published:** 2020-10-06

**Authors:** Irene Pertici, Giulio Bianchi, Lorenzo Bongini, Vincenzo Lombardi, Pasquale Bianco

**Affiliations:** PhysioLab, University of Florence, 50019 Sesto Fiorentino (FI), Italy; irene.pertici@unifi.it (I.P.); giuli.bianchi@student.unisi.it (G.B.); lnzbng@gmail.com (L.B.); pasquale.bianco@unifi.it (P.B.)

**Keywords:** myosin ensemble mechanics, myosin-based machines, synthetic nanomachines, dual laser optical tweezers

## Abstract

The emergent properties of the array arrangement of the molecular motor myosin II in the sarcomere of the striated muscle, the generation of steady force and shortening, can be studied in vitro with a synthetic nanomachine made of an ensemble of eight heavy-meromyosin (HMM) fragments of myosin from rabbit psoas muscle, carried on a piezoelectric nanopositioner and brought to interact with a properly oriented actin filament attached via gelsolin (a Ca^2+^-regulated actin binding protein) to a bead trapped by dual laser optical tweezers. However, the application of the original version of the nanomachine to investigate the Ca^2+^-dependent regulation mechanisms of the other sarcomeric (regulatory or cytoskeleton) proteins, adding them one at a time, was prevented by the impossibility to preserve [Ca^2+^] as a free parameter. Here, the nanomachine is implemented by assembling the bead-attached actin filament with the Ca^2+^-insensitive gelsolin fragment TL40. The performance of the nanomachine is determined both in the absence and in the presence of Ca^2+^ (0.1 mM, the concentration required for actin attachment to the bead with gelsolin). The nanomachine exhibits a maximum power output of 5.4 aW, independently of [Ca^2+^], opening the possibility for future studies of the Ca^2+^-dependent function/dysfunction of regulatory and cytoskeletal proteins.

## 1. Introduction

In the sarcomere, the structural unit of the striated (cardiac and skeletal) muscle, two antiparallel arrays of the myosin II motor domains extending from the thick filament pull the nearby thin, actin-containing filaments from the opposite extremities of the sarcomere toward the center (Figure 1a). The individual myosin motors interact with actin only briefly, generating piconewton force and nanometer displacement of the actin filament through an ATP-driven working stroke [1,2], and remain detached throughout most of their ATP hydrolysis cycle [3]. The production of steady force and shortening is an emergent property of the collective motor generated in each half-thick filament by the mechanical coupling between myosin motors via their attachment to the filament backbone. Cell studies have demonstrated that, in the intact cell environment, the half-sarcomere is able to efficiently work across a wide range of externally applied loads by tuning the number of heads attached to actin in proportion to the global filament load [4]. 

Both thin and thick filaments are involved in the regulation of the contraction of striated muscle. The classical thin filament-mediated mechanism implies that, following the rise of [Ca^2+^] induced by cell membrane depolarization, Ca^2+^ binding to troponin in the thin filament induces a structural change in the regulatory complex (troponin and tropomyosin) (Figure 1b) that makes the actin sites available for binding of the myosin motors [5]. Yet myosin motors in the resting muscle lie on the surface of the thick filament in the OFF state that prevents ATP hydrolysis and actin attachment [6,7]. A stress-sensing mechanism in the thick filament that switches motors ON to make them available for actin interaction has been recently characterized by combined mechanical and X-ray diffraction studies on intact muscle cells [8,9], but the molecular details of the mechanism and its control by accessory and cytoskeleton proteins (MyBP-C and titin, Figure 1b) are still poorly understood. 

Cell studies, on the other hand, are complicated by the large ensemble of motor proteins and filaments and by the confounding contribution of the sarcomeric (regulatory, accessory and cytoskeleton) proteins. Single-molecule mechanics use purified proteins and thus allow for the definition of the pure motor protein action and the comparison between different isoforms or native and mutant or genetically modified proteins. Single-molecule mechanics, however, suffer from the intrinsic limit in that they cannot investigate the properties that uniquely emerge from the collective motor within the architecture of the half-sarcomere. 

These limits have been overcome with the realization of a one-dimensional nanomachine titrated to contain the minimum number of motor molecules, which, upon interaction with an actin filament, are able to mimic the half-sarcomere performance, generating steady force and constant velocity shortening [10]. In the nanomachine, an ensemble of myosin II molecules, purified from the skeletal muscle, are carried on a piezo-nanomanipulator and brought to interact with an actin filament attached with the proper polarity to a bead (bead-tailed actin, BTA [11]) trapped on the focus of dual laser optical tweezers (DLOT, [12]). In perspective, the nanomachine has the possibility to selectively test the role of the other sarcomeric proteins by integrating them in the system one at a time. A main limit to this end is that to obtain correctly oriented BTA via gelsolin, a Ca^2+^-regulated actin-binding protein, [Ca^2+^] of 0.1 mM (corresponding to [free Ca^2+^] of 0.077 mM in the given experimental conditions [13]) is necessary [11]. Setting up an alternative method that preserves [Ca^2+^] as a free parameter is compelling for future investigations of the role of any other sarcomeric protein acting through Ca^2+^-dependent mechanisms. Among them, the most important is the Ca^2+^-dependent thin filament activation by the regulatory complex troponin–tropomyosin. The problem is overcome here by exploiting the recent availability of a Ca^2+^-insensitive gelsolin fragment (TL40 from Hypermol, see Materials and Methods). 

The force developed under isometric conditions and the force–velocity (*F-V*) relation of the nanomachine powered by eight HMMfragments of myosin purified from rabbit psoas in 2 mM ATP are determined either in the absence or in the presence of 0.1 mM Ca^2+^ (the concentration required for BTA preparation with gelsolin). At the same temperature (~23 °C), all the relevant mechanical parameters (development of the maximum force in isometric conditions (*F*_0_), maximum shortening velocity (the unloaded shortening velocity *V*_0_) and maximum power (*P*_max_)) are the same independent of the [Ca^2+^] and the protocol used for BTA assemblage. Moreover, a kinetic model simulation based on the performance of fast mammalian skeletal muscle is able to fit all the relevant mechanical parameters of the nanomachine, once the methodological limits of the in vitro mechanics are taken into account, simply scaling down the number of motors available for actin interaction. The nanomachine with BTA assembled using the Ca^2+^-insensitive gelsolin fragment TL40 preserves [Ca^2+^] as a free parameter, and the development of a multiscale model for interfacing data at different hierarchical levels of the striated muscle opens new avenues for the investigation of the Ca^2+^-dependent function/dysfunction of any specific sarcomere protein.

## 2. Results

### 2.1. A Version of the Nanomachine Independent of the Presence/Absence of Ca^2+^

The mechanical output of the synthetic nanomachine powered by myosin II from rabbit psoas in solution with 2 mM ATP was measured either in the presence of Ca^2+^ (0.1 mM CaCl_2_, BTA prepared with gelsolin) or in Ca^2+^-free solution (BTA prepared with the gelsolin fragment TL40), as described in the Materials and Methods (see also [10]) (Figure 2). The ensemble of HMM fragments of myosin, carried on a piezo-nanomanipulator acting as a length transducer, was brought to interact with an actin filament attached with the proper polarity to the bead trapped on the focus of dual laser optical tweezers (DLOT) acting as a force transducer (Figure 2a). 

The experiment in Figure 2b, done in Ca^2+^-free solution with the actin filament attached to the bead via the gelsolin fragment TL40, shows that in position clamp (time interval, *i*_t_ 1), following the formation of the actin–myosin interface, the force rises to a maximum steady value (*F*_0_) of ~ 16 pN. At this point, the control is switched to force (*i*_t_ 2) and then a staircase of force drops to 11 (*i*_t_ 3), 7 (*i*_t_ 4) and 3 pN (*i*_t_ 5) is imposed, which elicits shortenings at constant velocity progressively larger as force decreases. When the control is switched back to position clamp (*i*_t_ 6), the force recovers a maximum steady value identical to the initial *F*_0_, demonstrating that the number of motors available for attachment to the actin filament remains the same throughout the whole interaction.

As shown in Figure 2c, independent of the presence (dark gray) or the absence of Ca^2+^ (light gray dashed bars), the frequency distributions of *F*_0_ almost superimpose and the mean values are 15.41 ± 0.73 pN in 0.1 mM CaCl_2_ and 15.88 ± 0.66 pN in Ca^2+^-free solution. The difference is not significant according to the *t*-test (*p* > 0.1). Thus, there is no significant effect of Ca^2+^ on the isometric force. 

The *F-V* relation in either condition is plotted in Figure 2D (dark gray symbols in 0.1 CaCl_2_, light gray symbols in Ca^2+^-free solution), showing that the *F-V* points lie along the same relation irrespective of the presence/absence of Ca^2+^. The dashed lines are the fits to the respective *F-V* data using Hill’s hyperbolic equation [14]:(*F* + *a*) ⋅ (*V* + *b*) = (*F*_0_ + *a*) ⋅ *b*(1)
where a and b are the distances of the asymptotes from the ordinate and abscissa, respectively. In 0.1 mM CaCl_2_ experiments (dark gray dashed line), *V*_0_ (= *b*⋅*F*_0_/*a*, the ordinate axis intercept that estimates the maximum shortening velocity or the velocity of unloaded shortening) and *a*/*F*_0_ (the distance from the vertical asymptote to the ordinate axis, which gives an estimate of the curvature of the *F-V* relation) are 3.41 ± 0.33 µm s^−1^ and 0.25 ± 0.08, respectively, practically the same as *V*_0_ and *a*/*F*_0_ in Ca^2+^-free experiments (light gray dashed line), 3.56 ± 0.42 µm s^−1^ and 0.24 ± 0.09, respectively. In either case, the *t*-test for the differences gives *p* > 0.1. *V*_0_ is ~ 35% higher than the corresponding *V*_f_ (filled symbols, 2.56 ± 0.43 µm s^−1^ in Ca^2+^-free conditions and 2.43 ± 0.41 µm s^−1^ in solution with 0.1 mM CaCl_2_), the velocity of filament sliding on rabbit HMM in the in vitro motility assay (IVMA). According to the *t*-test, the difference is significant with 0.1> *p* > 0.05. A larger difference was previously observed between *V*_f_ determined in an IVMA with myosin extracted from frog muscle and *V*_0_ determined in situ in single fibers from the same frog muscle [15] and was attributed to the presence, in the IVMA, of non-specific interactions between actin filaments and the IVMA surface. Putatively, these interactions are less effective in the nanomachine than in the IVMA, suggesting that motors interacting with the actin filament are better exposed to interaction because they lie on a convex surface (Figure 2a and see Materials and Methods). 

The power–force (*P-F*) relations calculated from the *F-V* hyperbolic fits are plotted in Figure 2e and show a *P*_max_ value of 5.35 aW and 5.51 aW, in 0.1 mM CaCl_2_ and Ca^2+^-free solution, respectively. Thus, as expected from the curvature (defined by the parameter *a/F*_0_), P_max_ is also not affected by Ca^2+^. These results demonstrate that the performances of the nanomachine are identical when assembling the BTA with either gelsolin that implies 0.1 mM [Ca^2+^] or the gelsolin fragment TL40 in Ca^2+^-free solution. The use of the TL40 fragment totally preserves the nanomachine performance while making [Ca^2+^] a free parameter, opening the possibility for studies of the Ca^2+^-dependent mechanisms of the other sarcomere proteins. 

### 2.2. Interfacing In Vitro and In Situ Performance of Myosin II by Model Simulation

The power of the nanomachine to define the emergent properties of the half-sarcomere of striated muscle is tested by applying the same mechano-kinetic model of the actin–myosin interaction to fit the performances of both the nanomachine powered by myosin II from rabbit psoas muscle and the intact fast mammalian muscle. As shown in figure 3a (from [10]), the model assumes one detached state (D) and two different force-generating attached states (A1 and A2). The kinetics of state transitions are firstly constrained to fit the mechanics and energetics of the skeletal muscle. In the absence of a direct estimate of the mechanical parameters of the intact fast rabbit muscle, the muscle of reference that provides adequate mechanical and energetic information to be used as constraints for interfacing with in vitro results is the extensor digitorum longus (EDL) from the rat ([16] and references therein). This may imply a 10% overestimation of the kinetic parameters used in the simulation of rabbit muscle, due to the smaller size of the rat, on the basis of the size effect on ortholog myosin isoforms [17]; however, for reasons of simplicity, we did not apply any correction in adapting the rat muscle parameters to the rabbit myosin nanomachine. 

The mechanical constraints from whole muscle experiments at the temperature of the nanomachine experiments (23°C), already described in detail in the previous paper on the nanomachine [10], are: (*i*) isometric tetanic force *F*_0_ 250 kPa, (*ii*) unloaded shortening velocity *V*_0_ 8.6 μm s^−1^ per half-sarcomere and (*iii*) *F-V* relation as in Figure 3b, in which the data from [16] and references therein are fitted with Hill’s hyperbolic equation (dashed line), which gives a value of *a/F*_0_ (the parameter that estimates the curvature of the relation) of 0.36. The density of force per cross-sectional area can be translated to force per half-thick filament (htf), taking into account the thick filament density (~5.8.10^14^ m^−2^, [18] and references therein) and corresponds to 460 pN per htf (see abscissa intercept of the dashed line in Figure 3b). The power–force (*P-F*) relation, calculated from the *F-V* relation, shows a maximum power (*P*_max_) of 460 aW per htf attained at ~1/3 *F*_0_ (Figure 3c, dashed line).

Model simulations of mechanics and energetics of the muscle are obtained by adjusting the rate functions governing the kinetics of state transitions in the cycle of Figure 3a, as explained in detail in the previous paper [10]. Rate functions in Appendix A are reported from [10] (supplementary figure 6). Data for whole muscle in Figure 3b,c and Table 1 are reported from supplementary figure 6f,g and table 1 in [10]. How the most relevant energetic features are constrained by data in the literature is explained here. In isometric contraction, the rate-limiting step in the cycle is detachment from A2; under these conditions, the fraction of motors attached (the duty ratio) is the maximum, while the rate of ATP splitting per myosin head (*φ*_0_) is the minimum (~11 s^−1^ at room temperature in the mammalian muscle ([19] and references therein)). During steady shortening, the duty ratio decreases and the ATP splitting rate per myosin head (*φ*) increases, due to the increase in the rate of motor detachment following the execution of the working stroke [20,21,22,23]. However, *φ* at the load for the maximum power (*φ*_Pmax_) is higher than *φ*_0_ by no more than threefold [19], so that the resulting maximum power can be predicted only by assuming that, during shortening, the attached myosin motors can rapidly regenerate the working stroke by slipping to the next actin farther from the center of the sarcomere during the same ATPase cycle [21,23,24] (step “slip” in Figure 3a) and undergoing A1’-A2’ state equilibration according to step 2 kinetics [1]. Detachment from either A1’ or A2’ (step 3’) implies ATP hydrolysis. As shown in Figure 3b,c and in Table 1, the relations and the underlying mechanical and energetic parameters predicted by the model for the half-sarcomere of the mammalian muscle agree with those derived from published data [16,19]. Notably, *φ*_0_ calculated by the flux through step 1 at *F*_0_, is 11.6 s^−1^. At *P*_max_, *φ*_Pmax_ increases to 35.5 s^−1^ in the mammalian muscle (~3*φ*_0_).

The nanomachine performances, represented by open circles and dashed lines in Figure 3d and by the dashed line in e, where the *F-V* and *P-F* relations are obtained by pooling data in the absence and the presence of Ca^2+^ from Figure 2, are simulated after taking into account the effects of the methodological limits of the in vitro system with respect to muscle: (*i*) the compliance in series with the system constituted by the motor array and the overlapping bead-attached actin filament is two orders of magnitude larger than that of a muscle cell [10] due to the trap compliance, 3.7 nm pN^−1^; (*ii*) evidence from single molecule mechanics indicate that the random orientation of the motors on their support reduces the force developed at high load from the value exerted by the correctly oriented motor (*F*_c_) down to a minimum value of 0.1*F*_c_ attained when the motors are 180° away from the correct orientation [25]; (*iii*) similarly, single molecule mechanics showed that the step size (*L*) generated at low load by a motor lying on a cofilament (made of rods and myosin molecules) is reduced with respect to that (*L*_c_ ~10 nm) of the motor interacting with a correctly oriented actin filament, when the actin filament is oriented in the opposite direction [26]. These constraints are introduced in the model by reducing, progressively with the deviation from the correct orientation (0°), the isometric force and the shortening range for which an attached force generating motor (A2 state) maintains force, according to the family of A2 force functions shown in Appendix A (light red lines). The relevant parameters describing the force function range from *F*_c_ = 5 pN and *L*_c_ = 10 nm, exhibited by a correctly oriented motor (black line from supplementary figure 6b in [10]), to 0.5 pN and 1.5 nm, respectively (red line), exhibited by a motor 180° away from the correct orientation. Correspondingly, the slope of the A1 force function, representing the stiffness of the attached motor, scales from 2 pN nm^-1^ (black dashed line) to 0.2 pN nm^−1^ (red dashed line). This is a quite sensible assumption, considering that for a motor away from the correct orientation the S2 domain can also contribute to the compliance [27,28]. 

Under these conditions, the model is able to fit all the features of the *F-V* and *P-F* relations of the myosin-based nanomachine (continuous lines in Figure 3d,e, respectively), using as the only free parameter the number of motors available for the actin interaction (*N*), which must be scaled down from that in the half-thick filament, 294 [29], to that in the nanomachine, 16, in the experiments under consideration [10]. This is shown in detail in Figure 3f, in which the horizontal dashed lines, representing the experimental values for *F*_0_ (16.79 ± 0.38 pN), *P*_max_ (5.45 aW) and *V*_0_ (3.54 ± 0.13 µm s^−1^), intersect the relations between the corresponding simulated parameters and *N* (dots) for *N* = 16. The value of 16 is twice the number of rupture events recorded from the ensemble of motors deposited on the same support in ATP-free (rigor) conditions (see figure 1 in [10]). Assuming that each motor of the myosin dimers identified as available for actin interaction by rupture events in rigor behaves independently in 2 mM ATP, eight rupture events in rigor imply *N* = 16, in agreement with the prediction of the model simulation. It is worth noting that the model predicts that *F*_0_ and *P*_max_ rise continuously with *N*, while *V*_0_ attains a plateau for N ≥ 24. In fact, according to the AF Huxley 57 model [20], *V*_0_, the unloaded shortening velocity, is attained at a velocity at which a balance is attained between motors undergoing the working stroke and promoting sliding and motors at the end of the working stroke and resisting sliding. Therefore, this *V*_0_ is intrinsically independent of the number of available motors and solely dependent on the rate constant for motor detachment. The simulation predicts that this condition is no longer realized if *N* is <24, suggesting that below a given number of motors in the ensemble, the reduction of the duty ratio with the increase in shortening velocity reduces the number of motors attached at any time to less than three, and there is a consequent loss of the conditions for continuous interaction [30] which appears as a reduction in the average sliding velocity [10].

The application of the model simulation to the nanomachine implies, beyond the depressant effect of the random orientation of motors on both the step size and force of the motor (Appendix A), the introduction of a one hundred times larger in-series compliance, which causes the strain-dependent kinetics of the attached motors to generate the push–pull experienced by the motors when actin slides away from–toward the bead for the addition–subtraction of the force contribution by each motor, as described in detail in [10]. As a consequence, the model predicts that, with respect to the values in the corresponding muscle, *φ*_0_ increases by 70% (Table 1). At *P*_max_, the model predicts that the nanomachine undergoes an increase in *φ*_Pmax_ smaller than that of the muscle (Table 1) as a consequence of the depression in the step size and thus in the low-load velocity of shortening, due to the random orientation of motors.

## 3. Discussion

A synthetic nanomachine, made by an array of eight myosin dimers of myosin II from the fast skeletal muscle of rabbit interacting with an actin filament, is used to mimic the emergent properties of the half-sarcomere of striated muscle, namely the generation of steady force and constant velocity shortening (and thus steady power), in the absence of the effects of the other sarcomere (regulatory, accessory and cytoskeleton) proteins.

The unique performance of our system emerges clearly from the comparison with the existing alternative approaches, which failed to produce steady force and power for the action of a preset number of motors, preventing any interpretation in terms of the kinetics and energetics of the actin–myosin interaction in relation to the load [35,36,37,38]. The limits intrinsic in our design, consisting in reduced force and step size generated by randomly oriented motors, could be overcome by motor systems such as the native isolated thick filament or the synthetic cofilament made of rods and myosin molecules [39]. However, the performance of the systems based on single actin–single myosin filament [39,40,41] appears systematically limited to transient displacements of the actin filament abruptly interrupted after variable time periods, failing to produce steady force and power by a constant number of interacting motors. The reason for the success of our design is the possibility to ensure that the array of interacting motors and the actin filament lie on parallel planes, in this way preserving the in situ motor condition of being “in-parallel force generators”. The lack of alignment between the actin filament and the motor ensemble in all the other existing systems is in turn the reason for their limited performance, as the attached head closest to the bead experiences an additional stress generated by the out-of-axis vertical component of the ensemble force. 

In perspective, the nanomachine allows for the role of the sarcomeric (accessory and regulatory) proteins to be investigated by integrating them in the system one at a time, but in the original system [10], this perspective had a main limit in the use of gelsolin, a Ca^2+^-regulated actin-binding protein, for attaching properly oriented actin filaments to the bead (BTA) trapped in the focus of the DLOT used as a force transducer. In fact, future investigation of the role of any other sarcomeric protein acting through Ca^2+^-dependent mechanisms requires the preservation of [Ca^2+^] as a free parameter. The problem is solved here by devising a new protocol in which the BTA is prepared using the Ca^2+^-insensitive gelsolin fragment TL40. The effect of Ca^2+^ on the performance of the nanomachine in its simplest version (eight HMM fragments from the fast isoform of rabbit myosin II pulling on an actin filament) is tested, demonstrating that none of the relevant mechanical parameters, the isometric force *F*_0_ (16.8 pN), the unloaded shortening velocity *V*_0_ (3.5 μm s^−1^) or the maximum power *P*_max_ (5.5 aW), depends on the presence/absence of Ca^2+^. Thus, the use of the TL40 fragment to prepare the BTA totally preserves the nanomachine performance while making [Ca^2+^] a free parameter, opening the possibility for studies of the Ca^2+^-dependent mechanisms of the other sarcomere proteins.

A simplified mechano-kinetic scheme (Figure 3a) is used first to fit the functional features of the fast mammalian skeletal muscle and then scaled down to the dimensions of the synthetic machine powered by eight HMM fragments from the rabbit psoas myosin. The bulk of mechanical and energetic data present in the literature of mammalian muscle [16,19] are taken into account to constrain the rate functions of state transitions (Appendix A) at the temperature (23 °C) of the nanomachine experiments.

The present version of the model is an implementation of that already published [10] in which the predicted *F-V* relation (dotted line in Appendix A from figure 4 of [10]), provided higher than observed shortening velocities in the region of very low loads. The reason for the discrepancy is the absence of consideration, in the previous model simulation, of the depressant effect on the step size of motors moving away from the correct orientation with respect to the actin filament [26]. In Appendix A, the family of light red continuous lines shows the effect, on the *d* dependence of the force of A2 motors, of the progressive deviation of motor orientation from the correct orientation (green continuous line) to the orientation in the opposite direction (red continuous line). It must be noted that the observed *F-V* relation (circles and dashed line in Figure 3d and Appendix A) is predicted by the model (continuous line) only if the values of both *d* for which the A2 motors maintain the isometric force and the abscissa intercept of the A2 force function are scaled down in proportion to the reduction of the isometric force (the ordinate intercept in Appendix A). Single-molecule experiments reporting a smaller effect of opposite orientation on the low-load step size than on the force [26] would suggest a reduced effect of wrong motor orientation on the abscissa intercept of the A2 force function. However, assuming a depressant effect on the step size (and a reduction in the abscissa intercept of the A2 force function) less than that shown by the family of curves in Appendix A shifts the simulated low-load *F-V* points above the observed one.

The evidence that a property of the motor array, like *V*_0_, can be predicted by adjusting a single molecule parameter, like the step size, is a remarkable demonstration of the efficiency of our approach as a tool for interfacing single motor properties with the properties emerging from its collective arrangement. Moreover, the implementation of a unique model able to fit in vitro data from the nanomachine and in situ data from the intact muscles, accounting for the methodological reasons that limit the nanomachine performance when it fails to get the expected muscle performance, generates an unprecedented two-way communicating path. From one side, the model allows the definition of the minimal conditions of the synthetic nanomachine able to reproduce the power and efficiency shown at the muscle level and, from the other, the model and the nanomachine integrate to provide a new powerful tool for future applications in which the performance of the synthetic machine powered by mutant and engineered myosins can be exploited to predict their outcome at the organ level. 

## 4. Materials and Methods 

### 4.1. Preparation of Proteins

Adult male rabbits (New Zealand white strain), provided by Envigo, were housed at Centro di servizi per la Stabulazione Animali da Laboratorio (CeSAL, University of Florence), under controlled conditions of temperature (20 ± 1 °C) and humidity (55 ± 10%), and were euthanized by injection of an overdose of sodium pentobarbitone (150 mg kg^−1^) in the marginal ear vein, in accordance with the Italian regulation on animal experimentation (Authorization 956/2015-PR) in compliance with Decreto Legislativo 26/2014 and EU directive 2010/63. Two rabbits were used for the experiments.

HMM fragment of myosin was prepared from rabbit psoas muscle as reported previously [10]. The functionality of the purified motors was always preliminarily checked with an in vitro motility assay.

Actin was prepared from leg muscles of the rabbits according to [42]. Polymerized F-actin was fluorescently labeled by incubating it overnight at 4 °C with an excess of phalloidin-tetramethyl rhodamine isothiocyanate [43]. 

The correct polarity of the actin filament, that is its attachment to the bead with the + end [10], was obtained in two ways: (*i*) in the conventional way [11], the +end of the filament was attached to a polystyrene bead (diameter 3 µm) with the Ca^2^-sensitive capping protein gelsolin, (*ii*) according to a new protocol, in which full-length gelsolin (subdomains S1-S6) was replaced by the Ca^2+^-insensitive gelsolin fragment TL40, consisting of the N-terminal homologous subdomains S1-S3 (Hypermol, Germany), so that the nanomachine could be assembled in the absence of Ca^2+^. In preliminary tests, Ca^2+^ was removed from the solutions at various stages of TL40-coated bead preparation, showing no differences in the ability and the specificity to bind the actin filament. Gelsolin-coated beads and TL40-coated beads were stored in stock solution (150 mM NaCl, 20 mM sodium phosphate buffer pH 7.4, 0.1 mM ATP, 10 mg ml^-1^ BSA, 5% (*v/v*) glycerol and 3 mM NaN_3_) at 0 °C for about 6 months.

### 4.2. Mechanical Measurements

The mechanical apparatus is described in detail in [10]. Briefly, the myosin motors, deposited on the functionalized surface of a chemically etched single-mode optical fiber carried on a three-way piezoelectric nanopositioner (nano-PDQ375, Mad City Lab, Madison WI, USA) acting as a displacement transducer, were brought to interact with an actin filament attached with the correct orientation to the bead (bead-tailed actin, BTA) trapped in the focus of dual laser optical tweezers (DLOT, [12]) acting as a force transducer. The force–displacement transducer system had a dynamic range for both force (0−200 pN, resolution 0.3 pN) and displacement (0−75,000 nm, resolution 1.6 pN) adequate to measure the output of the nanomachine and could be servo-controlled either in position or force feedback [10,44]. The frequency response of the system in position feedback was limited by the rise time of the piezo-stage movement (*t*_r_ ~2 ms), while in force feedback, it was also limited by the damping exerted on the bead by the viscosity of the medium, so that *t*_r_ was ~20 ms. Mechanical protocols are described in the Results section. For all the experiments, [HMM] was 100 μg mL^−1^ [10].

The nanomachine was hosted in a flow chamber which had two separate compartments in which actin and myosin were flowed separately, thereby preventing uncontrolled protein interactions: one compartment was used to introduce the bead-attached actin, the other compartment, in which the support for the myosin motors was mounted, was used to introduce HMM dissolved in buffer A (25 mM imidazole pH 7.4, 33 mM KCl, 0.1 mM CaCl_2_, 5 mM MgCl_2_, 10 mM DTT and 2 mM ATP) [10]. The support for the motors was the lateral surface of a single-mode optical fiber, chemically etched to a diameter of ~4 µm and functionalized for HMM attachment by coating it with nitrocellulose (1% *w/v*) [42]. The fiber was positioned in the HMM compartment just before the confluence of the two compartments. Actin-attached beads from the other compartment were trapped with the optical tweezers one by one at the intersection of the flows to select those with a single actin filament at least 6–7 μm long. Then the flow chamber was moved across the x-y planes by means of a micro-positioner carrying the nanopositioner to bring the bead-attached actin close to the support with the motor array. The final positioning was achieved under nanopositioner control. To start the experiment, buffer B (buffer A plus 0.1 mg mL^−1^ glucose oxidase, 20 µg mL^−1^ catalase, 5 mg mL^−1^ glucose and 0.5% *w/v* methylcellulose, 400 cP) was flowed at 10 µL min^−1^. The presence of methylcellulose inhibited the lateral diffusion of F-actin [30], thereby minimizing the probability that, in 2 mM ATP, the interaction terminated during low-force isotonic contractions or during the initial phases of force redevelopment following large releases. The 0.5% *w/v* methylcellulose did not affect the mechanical or kinetic properties of the nanomachine [10]. All experiments were conducted at room temperature (23 °C).

### 4.3. Data Analysis

The velocity of shortening (*V*, µm s^−1^) in the responses to the reduction of force to *F* values (pN) below the isometric value (*F*_0_) was measured by the slope of the displacement trace (the red trace interpolated with black lines for clarity in Figure 2b). The *F-V* data were fitted with the hyperbolic Hill’s equation [14]: (*F* + *a*) · (*V* + *b*) = (*V*_0_ + *b*) · *a,* where a and b are the distances of the asymptotes from the ordinate and abscissa, respectively, and V_0_ (the ordinate intercept) estimates the maximum or unloaded shortening velocity. Here, a is a parameter that is used to express the degree of curvature of the relation, has the dimension of a force (pN) and, when normalized for F_0_, is an index of the relative maximum power that can be delivered at intermediate forces [14]. The power output (P) at any force was calculated by the product between F and V and expressed as µm s^−1^ · pN = 10^−18^ W = aW. Dedicated programs written in LabVIEW (National Instruments) and Origin 2015 (OriginLab Corporation) were used for the analysis. Experimental points are reported as mean ± SEM unless otherwise stated.

### 4.4. Model Simulation 

The results were numerically simulated by a stochastic model already published [10] that estimates the probability distributions of potential results by allowing for random variation in inputs over time until the standard deviation of the result is lower than 5% of the mean value obtained from single cycles of 5 s iterations. The mechanical cycle of the motors is depicted in Figure 3a. The state transitions as well as the strain of the attached motors are stochastically determined according to the kinetics reported in Appendix A. Each attached motor exerts, on the actin filament, a force that depends on its conformation and its position with respect to the actin monomer to which it is attached. The model operates in either position or force clamp. The iteration time in the calculation of the dependent variable, Δ*t* (= 10^−5^ − 10^−6^ s), depends on the stiffness of the system, which is dominated by the trap compliance. Both the force generated by the motors, transmitted through the BTA, and the force of the optical trap act on the trapped bead. The motion of the bead was simulated with over-damped dynamics by using a drag coefficient calculated from the bead radius and the viscosity of the medium (Stokes’s law). 

## Figures and Tables

**Figure 1 ijms-21-07372-f001:**
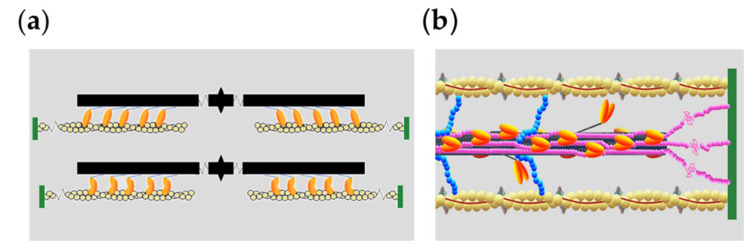
(**a**) Cartoon illustrating sarcomere shortening due to the reciprocal sliding between the myosin filament (black) originating from the center of the sarcomere (M line) and the actin filament (yellow) originating from the Z line at the end of the sarcomere (green), powered by the interdomain structural change in the S1 fragment of the myosin motor (orange) from the pre- (upper panel) to the post-working stroke conformation (lower panel). (**b**) Schematic representation of the half-sarcomere protein assembly at rest. Shown are actin (yellow), tropomyosin (Tm, red) and troponin complex (Tn, light and dark gray and brown) on the thin filament. On the thick filament (black), most of the S1 fragments of myosin dimers (orange) lie tilted back (OFF state) and the S1 fragments of two dimers move away with the tilting of their S2 rod-like domain (ON state); the MyBP-C (blue) lies on the thick filament with the C-terminus and extends to the thin filament with the N-terminus. Titin (pink) in the I-band connects the Z line to the tip of the thick filament and in the A-band runs on the surface of the thick filament up to the M line at the center of the sarcomere.

**Figure 2 ijms-21-07372-f002:**
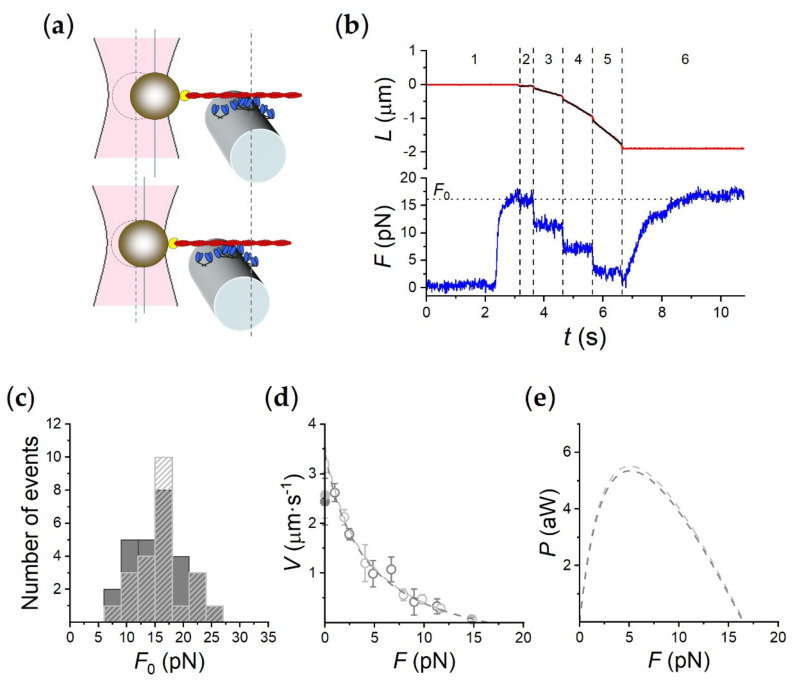
Performance of the rabbit HMM-based nanomachine. (**a**) Schematic representation of two snapshots during the interaction between the actin filament and the motor ensemble. Upper panel: in position clamp at *F*_0_; lower panel: in force clamp at 0.4 *F*_0_. (**b**) Recording of the actin filament sliding (*L*, upper trace, red) and force (*F*, lower trace, blue) during an interaction. Numbers bounded by dashed lines identify the different time intervals (*i*_t_) as detailed in the text. (**c**) Frequency distribution of *F*_0_. Data are plotted in classes of 3 pN; dark gray bars, measurements in 0.1 mM CaCl_2_ (= 77 µM free [Ca^2+^]); light gray dashed bars, in the absence of Ca^2+^. (**d**) *F-V* relation in 0.1 mM CaCl_2_ (dark gray open circles) and in Ca^2+^-free solution (light gray open circles). Points are mean ± SD from individual experiments, grouped in classes of force 0.15 *F*_0_ wide. Dark and light gray filled symbols on the ordinate are the *V*_f_ in the in vitro motility assay (IVMA) on rabbit HMM with and without Ca^2+^, respectively. The dashed lines are Hill’s hyperbolic equation fits to the data with the same color code as symbols. Data in **c** and **d** are from 28 experiments in 0.1 mM CaCl_2_ and 23 experiments in Ca^2+^-free solution. (**e**) Power (*P*) versus *F*, calculated from the corresponding *F-V* fits in (d), with the same color code.

**Figure 3 ijms-21-07372-f003:**
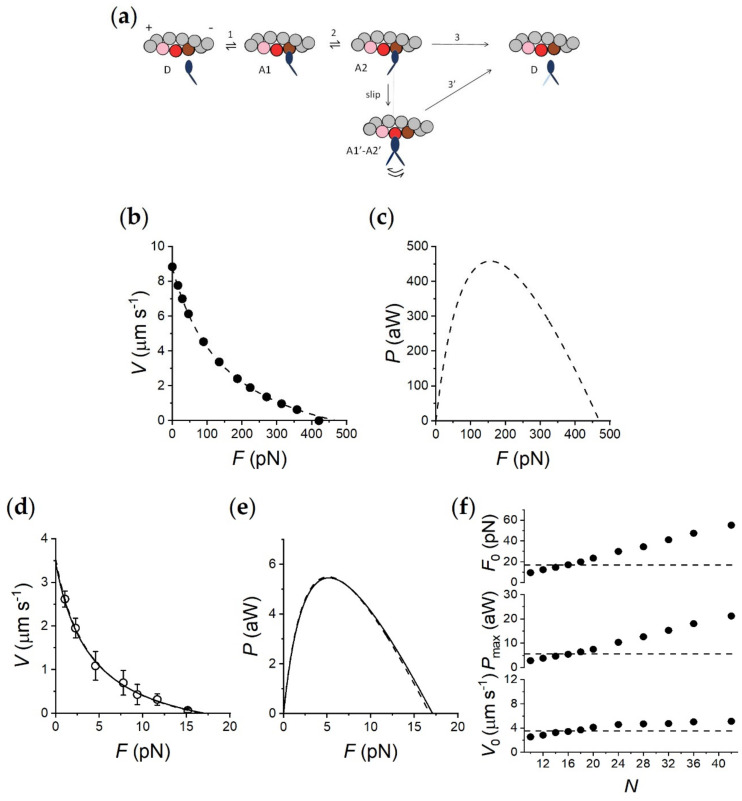
Model simulation. (**a**) Kinetic scheme from [10], with three states of the myosin motor (blue): D, detached; A1 and A2, attached to an actin monomer (brown). During shortening, the motor attached in the A2 state can slip to the next actin monomer farther from the center of the sarcomere (red) within the same ATPase cycle. The probability of a second slipping to the pink monomer is limited to 1/10 of that of the first slip. (**b**) Simulation of the *F*-*V* relation of fast mammalian muscle (adapted from supplementary figure 6f in [10]). The experimental *F*-*V* relation (filled circles) is calculated for the half-sarcomere at full overlap (294 motors available) from the data of fast mammalian muscle at room temperature [31,32]. The dashed line is Hill’s hyperbolic equation fitted to the results of the model simulation with a series compliance similar to that in situ (0.01 nm pN^-1^). (**c**) *P*-*F* relation calculated from Hill’s fit in b (adapted from supplementary figure 6g in [10]). (**d**) Experimental *F*-*V* relation obtained by pooling data in the absence and the presence of Ca^2+^ from Figure 2 (open circles) and fitted by Hill’s hyperbolic equation (dashed line), and its simulation calculated for a number of available heads *N* = 16 (continuous line). (**e**) Corresponding *P*-*F* relations. (**f**) Dependence on *N* of the three parameters featuring the machine performance, as indicated in the ordinate of each plot. The dashed lines indicate the respective experimental values.

**Table 1 ijms-21-07372-t001:** Simulated mechanical and energetic parameters of the muscle half-sarcomere and of the nanomachine. The parameters reported in the table are defined below and are accompanied by the references that provide the standard values used to constrain the simulation at the level of the muscle half-sarcomere. Data in the first line, concerning the whole muscle, are reported from Table 1 in [10]. *N,* number of available motors, which, per half-thick filament, are (49 crowns times 6 motors per crown =) 294 [29]; *F*_0_, isometric force referring to the half-thick filament [16]; *r*_0_, isometric duty ratio [33]; *φ*_0_, flux through step 1 of the cycle in Figure 3a in isometric conditions, corresponding to the ATP hydrolysis rate per myosin head at *F*_0_, [19] and references therein; *V*_0_, maximum shortening velocity [16]; *P*_max_, maximum power; *φ_P_*_max_, ATP hydrolysis rate per myosin head at *P*_max_ [19,34].

	*F*_0_ (pN)	*r*_0_	*φ*_0_ (s^−^^1^)	*V*_0_ (µm s^−^^1^)	*P*_max_ (aW)	*φ*_Pmax_ (s^−^^1^)
*N* = 294 Compliance 0.01 nm pN^−1^	433 ± 5	0.32	11.65	8.61 ± 0.16	462	35.50
*N* = 16 Compliance 3.7 nm pN^−1^ + random	15.8 ± 0.4	0.40	18.21	3.45 ± 0.13	5.45	26.21

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
