# Peer review of "A Myosin II-Based Nanomachine Devised for the Study of Ca2+-Dependent Mechanisms of Muscle Regulation"

_ijms, 2020, doi:10.3390/ijms21197372_

Round 1

Reviewer 1 Report

The authors attempted to reproduce steady force and shortening by the minimum number of motor molecules within the sarcomere-like architecture (nanomachine) in solution under physiological conditions. Some successful data were already published in 2018 (Nature Comm. Ref.10). Only improvement is an use of Ca-insensitive gelsolin instead of previous Ca-sensitive one for correcting the orientation of actin. This enables them to examine Ca regulation of muscle contraction using this nanomachine. The experiments were well conducted. However, the paper is not interesting because the results are the same as those in Ref.10. I raised only a few questions. Answers should be described in the text.

I do not understand why authors decided to use HMM rather than synthetic or natural myosin filaments, of which myosin molecules are correctly oriented on one-half side so that the effect of random distribution on the force generation can be avoided in the model simulation (the consideration in Supplementary Fig.2a).

The authors used 33 mM KCl. This is too low for physiological ionic strength 150-200 mM in muscle fibres. The biochemical study showed that lowering ionic strength greatly enhances the actin-myosin interaction in the presence of ATP. The results from nanomachine should not easily compared with those from intact muscle fibres. Please comment.

Oiwa et al. published the force-velocity relation of skeletal myosin at 1-2 mM ATP on Nitella-based actin cables (PNAS 86, 1510-1514 (1989); 87, 7893–7897 (1990)). The results were considerably different from that presented in this manuscript. Please comment.

Author Response

Response to Reviewer 1 Comments

We thank the reviewer for her/his comments that have suggested clarifications in the text, as detailed below

Point 1: The authors attempted to reproduce steady force and shortening by the minimum number of motor molecules within the sarcomere-like architecture (nanomachine) in solution under physiological conditions. Some successful data were already published in 2018 (Nature Comm. Ref.10). Only improvement is an use of Ca-insensitive gelsolin instead of previous Ca-sensitive one for correcting the orientation of actin. This enables them to examine Ca regulation of muscle contraction using this nanomachine. The experiments were well conducted. However, the paper is not interesting because the results are the same as those in Ref.10. I raised only a few questions. Answers should be described in the text.

I do not understand why authors decided to use HMM rather than synthetic or natural myosin filaments, of which myosin molecules are correctly oriented on one-half side so that the effect of random distribution on the force generation can be avoided in the model simulation (the consideration in Supplementary Fig.2a).

Response 1: The reasons that make our system unique, in comparison to any other existing system, for the ability of an ensemble of myosin motors to produce steady force and power and for the possibility of a quantitative analysis of the underlying kinetics and energetics were explained in detail in the first paper (Ref. 10). In particular it must be considered that any two-filaments system so far has failed to fit the conditions (1) for the motors to act as “in parallel” force generators, which requires that actin filament and myosin motors lie on parallel planes and (2) for the number of available motors to be defined and maintained constant during the interaction. The question is synthetically reported in the revised version (lines 299-313).  

Point 2: The authors used 33 mM KCl. This is too low for physiological ionic strength 150-200 mM in muscle fibres. The biochemical study showed that lowering ionic strength greatly enhances the actin-myosin interaction in the presence of ATP. The results from nanomachine should not easily compared with those from intact muscle fibres. Please comment.

Response 2: The reasons for selecting a ionic strength of 60 mM have been explained in previous work (Ref. 10 and 15). Namely, this ionic strength maximises the possibility to start actin-myosin interactions without affecting the strong actin-myosin interactions responsible for the nanomachine performance, while keeping  methylcellulose (that inhibits lateral diffusion of the action filament) as low as 0.5% (a concentration low enough to “not affect the mechanical and kinetic properties of the nanomachine”, lines 420-21).

Point 3: Oiwa et al. published the force-velocity relation of skeletal myosin at 1-2 mM ATP on Nitella-based actin cables (PNAS 86, 1510-1514 (1989); 87, 7893–7897 (1990)). The results were considerably different from that presented in this manuscript. Please comment.

Response 3: The question is more generally related to the comparison of the performance of our system with that of existing alternative systems and, as explained above, is now taken into consideration more explicitly at lines 299-313

Reviewer 2 Report

I have troubles with evaluation the manuscript and taking the decision.

Firstly, I am not really sure how much this manuscript differs from the reference 10, where they originally described the nanomachine (Pertici et al. 2018). I see here the same illustrations, some of the data are exactly the same (for example in table 1). Moreover, even the same “technology” with the gelsolin fragment has been described there.

I would expect that the authors clearly state the difference between both manuscripts and indicate the novelty of the current data.

Author Response

Response to Reviewer 2 Comments

We thank the reviewer for her/his comments that have been taken into account either for clarifying the text or in the reply, as detailed below

Point 1: I have troubles with evaluation the manuscript and taking the decision. Firstly, I am not really sure how much this manuscript differs from the reference 10, where they originally described the nanomachine (Pertici et al. 2018). I see here the same illustrations, some of the data are exactly the same (for example in table 1).

Response 1: We thank the reviewer for the request of clarification. To make an efficient evaluation of the advantages of the implementations introduced with respect to the previous paper (Ref 10) in relation to both methodology (attachment of correctly oriented actin filaments to the bead via the Ca2+ insensitive gelsolin fragment TL40) and modelling (introduction of the depressant effect of random orientation on the motor step size), we had to compare the nanomachine performance in the absence of Ca 2+ with that in the presence of Ca2+ (Fig 2).   Once demonstrated that the new method does not influence the performance, the pooled data are interpreted with the same kinetic scheme as data from whole muscle (Fig 3 and Table 1), for which data are necessarily the same as those shown in the previous paper. We now more systematically identify the data reported from previous paper both in the text (lines 192-194) and in the legends of Fig 3 (lines 272 and 276) and of Table1 (line 286).

Point 2: Moreover, even the same “technology” with the gelsolin fragment has been described there.

Response 2: The reviewer missed the fundamental point that in the previous paper the bead tailed actin was prepared with gelsolin which needs a [Ca2+] of 0.1 mM. The reason of the present paper is to obtain a nanomachine in which Ca 2+ is preserved as a free parameter for future studies of Ca2+ regulated mechanisms. We succeed in the task by demonstrating that the performance and the reliability of the nanomachine is preserved when the bead tailed actin is assembled with the Ca 2+ insensitive gelsolin fragment TL40.

Point 3: I would expect that the authors clearly state the difference between both manuscripts and indicate the novelty of the current data.

Response 3: Indeed, in Discussion we consider in detail the two main features that characterise the originality of this paper:

  1. The implementation of the preparation of bead tailed actin to preserve Ca2+ as a free parameter without affecting the nanomachine performance
  2. The refinement of the simulation taking into account the effect of random orientation on the step size. Consequently, we can demonstrate the power of the model simulation to interface single molecule properties (the step size) to ensemble properties (the velocity of shortening).

Reviewer 3 Report

This manuscript describes the use of a myosin II-based nanomachine, to mimic straited muscle and understand the effects of force and velocity generation without the influence of other proteins. Importantly, the research in this article elucidates both in vitro and in silico the force-velocity along with power-force relations in the absence and presence of Ca2+.

Overall, the results of the study demonstrate that the presence of Ca2+ has no significant impact on the TL40 gelsolin fragment and that allow for Ca2+ to be a free parameter for investigation of other Ca2+-dependent proteins in sarcomere. This study will help better understand mechanisms of other cytoskeletal proteins using the synthetic nanomachinery. I suggest the authors to take the following comments into consideration:           

Major Comments:

  1. In the article, the authors describe the use of nanomachine to understand the production of force and velocity by use of myosin and F-actin interactions. However, is the concentration of F-actin such that there is a network of filaments or are you using only 1 F-actin interactions with the coated beads? How would you know that the nanomachine is not binding to more than one filament? If this is that the beads are binding to more than one F-actin than would the force, velocity, or power produced be greater or reduced?
  2. In accordance, to the modeling and pooling of data from the experimental portions of the study, Figure. 3b and d, the data associated to simulation shows that the modeling predicts a larger F-V relation at lower force (pN) levels. However, when compared to the pooling of the experimental data the overall velocity produced is much lower when compared to the modeling of the same range. What variables would go into the modeling to account for why these values are so different?

Minor Comments:

  • Would the range of Ca2+ used in this study affect the structural and mechanical properties of actin filaments? This may have an effect on the force measurement or the sliding length of actin filaments. 
  • In the study, TL 40 gelsolin fragment was used would you expect different or similar results with the use of whole gelsolin protein? Please expand in the discussion if necessary.
  • In the results of Figure 2C, why are there larger number of events for no Ca2+ present? Is there another ion at play which could induce the rise in events taking place?
  • For the data associated to Figure 3F, why is there a plateau of velocity as the number of motors available increases? Is this a limitation for F-actin interaction? Possible number of allowable binding events? Please discuss.

Author Response

Response to Reviewer 3 Comments

We thank the reviewer for her/his comments that have suggested clarifications in the text, as detailed below

Point 1: This manuscript describes the use of a myosin II-based nanomachine, to mimic straited muscle and understand the effects of force and velocity generation without the influence of other proteins. Importantly, the research in this article elucidates both in vitro and in silico the force-velocity along with power-force relations in the absence and presence of Ca2+.

Overall, the results of the study demonstrate that the presence of Ca2+ has no significant impact on the TL40 gelsolin fragment and that allow for Ca2+ to be a free parameter for investigation of other Ca2+-dependent proteins in sarcomere. This study will help better understand mechanisms of other cytoskeletal proteins using the synthetic nanomachinery. I suggest the authors to take the following comments into consideration:           

Major Comments:

  1. In the article, the authors describe the use of nanomachine to understand the production of force and velocity by use of myosin and F-actin interactions. However, is the concentration of F-actin such that there is a network of filaments or are you using only 1 F-actin interactions with the coated beads? How would you know that the nanomachine is not binding to more than one filament? If this is that the beads are binding to more than one F-actin than would the force, velocity, or power produced be greater or reduced?

Response 1: The question of the reviewer about the possible presence of more than on actin filament on the bead is quite sensible. As explained in the text (present version lines 411-413), we selected beads, after trapping them, with only one filament and sufficiently long, the other were discarded. Beads with more than one filament had to be discarded because would provide artefactual larger values of isometric force and inconsistent F-V points. 

Point 2:

  1. In accordance, to the modeling and pooling of data from the experimental portions of the study, Figure. 3b and d, the data associated to simulation shows that the modeling predicts a larger F-V relation at lower force (pN) levels. However, when compared to the pooling of the experimental data the overall velocity produced is much lower when compared to the modeling of the same range. What variables would go into the modeling to account for why these values are so different?

Response 2: Data in Fig.3 b are (and c) are only for the whole muscle, either experimental (filled circles) or simulated (dashed line): the superposition shows the quality of the simulation. Data in Fig. 3d (and e) are only for the nanomachine, either experimental (open circles and dashed line) or simulated (continuous line). In this way we show the quality of the simulation of experimental data for either the whole muscle or the nanomachine made of 16 motors. Thus, the difference in velocity between the F-V relations in b and d are due to the differences present in the experimental data. Using the same kinetic scheme we show that these difference can be explained by the simulation once one takes into account the methodological limits of the nanomachine, namely the 100 times larger in series compliance and the random orientation of the myosin motors.

Point 3: Minor Comments:

Would the range of Ca2+ used in this study affect the structural and mechanical properties of actin filaments? This may have an effect on the force measurement or the sliding length of actin filaments.

Response 3: One of the aims of the work, made possible by the introduction of the gelsolin fragment TL40 for actin attachment to the bead, was to test whether there was some effect of Ca2+ on the performance of the basic version of the nanomachine, made only  by an actin filament and myosin motors. The results in Fig 2 shows no effects.

Point 4: In the study, TL 40 gelsolin fragment was used would you expect different or similar results with the use of whole gelsolin protein? Please expand in the discussion if necessary.

Response 4: In Fig. 2 the isometric force and the F-V relation obtained using the gelsolin fragment TL40 to attach the actin filament to the bead in the absence of Ca2+ (light gray) are compared with those obtained using gelsolin (which requires 0.1 mM Ca2+) (dark gray): the results are identical.

Point 5: In the results of Figure 2c, why are there larger number of events for no Ca2+ present? Is there another ion at play which could induce the rise in events taking place?

Response 5: The number of experiments are 28 in 0.1 mM Ca and 23 in Ca-free different by chance. In either case we considered the number adequate for the definition of both the isometric force and the F-V relation.

Point 6: For the data associated to Figure 3F, why is there a plateau of velocity as the number of motors available increases? Is this a limitation for F-actin interaction? Possible number of allowable binding events? Please discuss.

Response 6: The reviewer question let us to note that we missed to explain why the isometric force, the maximum power and the unloaded shortening velocity are expected to depend in different way on the number of available motors. This is now discussed at lines 246-255.  We thank to reviewer for promoting this clarification.

Round 2

Reviewer 2 Report

The authors have responded to my comments and some of the issues have been clarified so I can evaluate the manuscript.

Major comments

- the manuscript is definitely too long and lacks clarity. It should be substantially shorten and I believe this will make the message more clear.

- the manuscript could be aided by the EM presentation of the nanomachine

Minor comments,

- line 31, the author state that myosin II is extending from the thick filament while it is its fragment containing the head and S2. It has to be corrected.

- lines 53, please correct the grammar; mechanics in this case is singular.

Author Response

Response to Reviewer 2 Comments, 2nd revision

Point 1: The authors have responded to my comments and some of the issues have been clarified so I can evaluate the manuscript.

Major comments

- the manuscript is definitely too long and lacks clarity. It should be substantially shortened, and I believe this will make the message clearer.

Response 1: This request of the reviewer was not present in the previous review, in which the relevant argument was that some Figures and procedures were the same as those in a previous paper (Pertici et al., Nat. Commun. 2018, Ref [10]). In the reply and in the revised MS we have further clarified the technological advancement and that the data reported from the previous paper were referring to the whole muscle (thus cannot be different in the present paper) and were necessary for a comparative evaluation of the original in vitro mechanical results in this paper. In the 2nd report the reviewer adds the new claim of shortening the paper, without precise justification: the reviewer is not suggesting what part of the paper can be suppressed without losing coherence. In any case, following the reviewer suggestion, we decided to suppress the detailed explanation of the chemomechanical cycle (30 lines from line 451 in the present version), which may be redundant as it was already described in [Ref 10], and to refer to that one. Eventually, please note that the length of the revised version had increased to satisfy precise questions of the other two reviewers (reviewer 1: lines 300-314), reviewer 3: lines 247-256).

Point 2: the manuscript could be aided by the EM presentation of the nanomachine

Response 2: It looks odd to request an EM of the nanomachine as a whole, which is a system interfacing 3D hard-matter made objects and proteins. On the other hand, assuming that the reviewer intended a picture of the motor ensemble, please note that AFM images of the support with and without the myosin motors were reported in detail in [10] (Supplementary Fig. 8).

Point 3: Minor comments,

- line 31, the author state that myosin II is extending from the thick filament while it is its fragment containing the head and S2. It has to be corrected.

Response 3: We changed to “myosin II motor domains extending” from the thick filament”. 

Point 4: - lines 53, please correct the grammar; mechanics in this case is singular.

Response 4: Mechanics as a branch of Physics is used with a singular verb. Here “Mechanics” stays for “apparatus(es) for single molecule mechanical recording”, which can be used either with singular or with plural verb, so the grammar is correct. In any case, following the request of the reviewer we changed to singular.

Round 3

Reviewer 2 Report

I do not have additional comments.